# NE-MTOC Formation in Skeletal Muscle Is Mbnl2-Dependent and Occurs in a Sequential and Gradual Manner

**DOI:** 10.3390/cells14040237

**Published:** 2025-02-07

**Authors:** Payel Das, Robert Becker, Silvia Vergarajauregui, Felix B. Engel

**Affiliations:** Experimental Renal and Cardiovascular Research, Department of Nephropathology, Institute of Pathology and Department of Cardiology, Friedrich-Alexander-Universität Erlangen-Nürnberg (FAU), 91054 Erlangen, Germany; payel.das101192@gmail.com (P.D.);

**Keywords:** Mbnl2, non-centrosomal microtubule-organizing center, MTOC, nuclear envelope, pericentrin, skeletal muscle, splicing

## Abstract

Non-centrosomal microtubule-organizing centers (ncMTOCs) are important for the function of differentiated cells. Yet, ncMTOCs are poorly understood. Previously, several components of the nuclear envelope (NE)-MTOC have been identified. However, the temporal localization of MTOC proteins and Golgi to the NE and factors controlling the switch from a centrosomal MTOC to a ncMTOC remain elusive. Here, we utilized the in vitro differentiation of C2C12 mouse myoblasts as a model system to study NE-MTOC formation. We find based on longitudinal co-immunofluorescence staining analyses that MTOC proteins are recruited in a sequential and gradual manner to the NE. AKAP9 localizes with the Golgi to the NE after the recruitment of MTOC proteins. Moreover, siRNA-mediated depletion experiments revealed that Mbnl2 is required for proper NE-MTOC formation by regulating the expression levels of *AKAP6β*. Finally, Mbnl2 depletion affects *Pcnt* isoform expression. Taken together, our results shed light on how mammals post-transcriptionally control the switch from a centrosomal MTOC to an NE-MTOC and identify Mbnl2 as a novel modulator of ncMTOCs in skeletal muscle cells.

## 1. Introduction

Cells need to control their shape and internal organization in order to carry out essential functions such as division and movement. For this purpose, cells utilize several filamentous proteins that together form the so-called cytoskeleton. The eukaryotic cytoskeleton is composed of three filaments: (i) actin-based microfilaments, which help to maintain cell shape and mediate locomotion; (ii) intermediate filaments, which provide mechanical stability to the cells; (iii) microtubules, which determine the position of membrane-enclosed organelles, direct intracellular transport, and help in chromosome segregation during cell division via mitotic spindle formation [1]. Notably, the organization of the cytoskeleton changes markedly during differentiation to enable mammalian cells to execute highly specific functions. For example, the centrosome represents the dominant microtubule-organizing center (MTOC) in proliferating cells [2]. Yet, upon differentiation, MTOC function is assigned in various cell types to non-centrosomal sites [3,4], such as axons and dendrites of neurons [5], striated muscle cells [6], and osteoclasts [7,8], as well as at epithelial cells [9].

ncMTOCs are poorly understood; until recently, it was not known how ncMTOC formation is initiated or regulated, and the composition and function of ncMTOCs were elusive. Recently, we elucidated several of these processes by studying the NE-MTOC in striated muscle and osteoclasts. We have identified the composition of the NE-MTOC in cardiomyocytes [8], identified AKAP6 as a key organizer of NE-MTOC formation [6,8], showed that the transcription factor myogenin controls NE-MTOC formation in skeletal muscle via AKAP6 [6], and found that ectopic myogenin expression is sufficient to induce NE-MTOC formation in fibroblasts [6]. Finally, we showed that the NE-MTOC in cardiomyocytes is required for hypertrophic growth [8], in osteoclasts for bone resorption [8], and in skeletal muscle for nuclear positioning [6]. Thus, a deeper understanding of ncMTOC formation holds great promise for modulating the function of differentiated cells—for example, to reduce pathological hypertrophy [10] or bone loss in rheumatoid arthritis. The importance of myonuclear positioning has been illustrated by a number of muscle diseases that feature mislocalized nuclei [11,12]. Therefore, we addressed here the question of how the components of the NE-MTOC are assembled and whether post-transcriptional mechanisms that control NE-MTOC formation exist. There are several data available indicating that NE anchor platform proteins, MTOC proteins, and Golgi are recruited to the NE in a controlled spatiotemporal pattern. In cardiomyocytes, NE-MTOC formation starts after embryonic day 15, with AKAP6 and PCM1 being the first to be detected at the NE [8]. Perinuclear distribution of Pcnt and CDK5RAP2 is seen only after birth [13]. In C2C12 mouse myoblasts, the translocation of PCM1, Pcnt, and CDK5RAP2 to the NE occurs before myoblast fusion [14]. Notably, it has been reported that PCM1 and Pcnt are recruited to the NE in a gradual manner from a centrosomal localization [6].

Upregulation or switching of protein isoforms might be a general mechanism of NE-MTOC formation. Notably, several important components of the NE-MTOC are protein isoforms specifically upregulated during differentiation, such as the two key proteins, AKAP6β [8,15] and nesprin-1α [16], which are sufficient in a heterologous system to recruit endogenous centrosomal proteins to the NE [8]. Another example of an isoform switch is the expression of two Pcnt isoforms in cardiomyocytes: Pcnt S, which localizes to the NE, and Pcnt B, the dominant isoform at the centrosome [13]. Furthermore, alternative splicing of MTOC proteins has been described in other tissues. For example, while ninein has a centrosomal localization in neural progenitor cells, a splicing isoform switch resulting in a non-centrosomal localization is required for neuronal differentiation [17]. Further evidence comes from Drosophila studies, where a spermatid-specific isoform of centrosomin, an orthologue of CDK5RAP2, can induce ncMTOC formation at mitochondria [18].

The process of alternative splicing is regulated by RNA-binding proteins that bind to sequence-specific elements in pre-mRNAs and promote either the inclusion or exclusion of alternative exons in the mRNA [19]. The Muscleblind (Mbnl) protein family is one of the most enriched muscle-specific splicing factor families [20], and thus, its family members are good candidates for being regulators of NE-MTOC formation. In vertebrates, three Mbnl paralogs are known: Mbnl1, Mbnl2, and Mbnl3. These isoforms are encoded by three separate genes that map to chromosomes 3, 13, and X, respectively. They share their amino termini but differ in their C-termini [20,21]. In humans and mice, Mbnl1 expression is observed to be the highest in skeletal muscle and the heart, while Mbnl2 shows a more uniform distribution across various adult tissues, and Mbnl3 is highly expressed in the placenta [20,22]. The Mbnl proteins are Cys3His zinc-finger-containing proteins, originally identified in Drosophila as a critical factor for the terminal differentiation of photoreceptors and skeletal muscle [21,23]. Previous studies have shown that Mbnl1 is the dominant splicing factor in skeletal muscle [24]. Mbnl2 is primarily responsible for other RNA metabolic processes such as mRNA transport and stability [25,26], while Mbnl3 is inhibitory to myogenesis [27,28].

Mbnl proteins gained their initial importance because of their pathogenic role in a neuromuscular disorder, myotonic dystrophy (DM) [29,30]. DM type 1 (DM1) is characterized by an expansion of a CUG repeat in the 3′-UTR of the *DMPK* gene, and type 2 (DM2) is characterized by an expansion of a CCUG repeat in the first intron of the *ZNF9* gene. The expanded transcripts are retained in the nucleus as ribonuclear foci and sequester Mbnl proteins into the nucleus, leading to a reduction in the level of Mbnl proteins at their physiological locations and contributing to the pathogenesis of the disease [29,30,31]. Very little is, however, known about the molecular nature of Mbnl protein functions besides alternative splicing. Furthermore, although there have been advances in understanding the role of Mbnl proteins during the terminal differentiation of skeletal muscle, their role in early differentiation, the time when NE-MTOCs are established, has not yet been investigated. Besides acting as splicing factors, recent studies in cancer cells have indicated that Mbnl proteins can participate in regulating mRNA stabilization and transport of several genes. For example, Mbnl1 is reported to play a role in stabilizing metastatic suppressors such as *Snail*, *DBNL*, and *TACC1*, thereby preventing cancer metastasis [32,33]. Mbnl2 acts as a tumor suppressor by stabilizing *p21* mRNA [25]. Furthermore, Mbnl2 has been shown to be specifically upregulated in lung carcinoma cell lines, where it directly interacts with the mRNA of adhesion molecule *integrin α3* to facilitate its transport from the cell body to adhesion plaques. In this study, we used in vitro differentiation of C2C12 mouse myoblasts to study NE-MTOC formation and showed that MTOC proteins are recruited to the NE in a sequential and gradual manner, indicating the appearance of two proteins one after the other at the NE or the appearance of a protein at the NE that gradually, over time, covers it. Furthermore, we reveal through siRNA experiments that Mbnl2 regulates NE-MTOC formation, most likely by controlling *AKAP6β* expression and *Pcnt* splicing.

## 2. Materials and Methods

### 2.1. C2C12 Myoblast Culture and Differentiation

C2C12 cells (ATCC (Manassas, VA, USA), CRL-1772, RRID: CVCL_0188) were maintained and differentiated as previously described [6].

### 2.2. RNA Interference

Knockdown of gene expression was achieved through RNA interference utilizing a liposome-based transfection method as previously described [6]. Small interfering RNAs (siRNAs, 10 µM stocks, Integrated DNA Technologies, B.V. Leuven, Belgium) were used at a concentration of 40 nM (in the case of double depletion: 20 nM for siMbnl1 and 20 nM for siMbnl2) and mixed with either 2 µL (per well of a 24-well plate) or 8 µL (per well of a 6-well plate) of Lipofectamine RNAiMax (Life Technologies GmH (Darmstadt, Germany)) diluted in Opti-MEM (Life Technologies GmH (Darmstadt, Germany)).

### 2.3. RNA Isolation and cDNA Preparation

RNA from C2C12 cells was isolated using a column-based RNA purification kit (Macherey-Nagel GmbH & Co. KG (Düren, Germany)) according to the manufacturer’s protocol. Reverse transcription of RNA into cDNA was carried out in two steps. In the first step, 1 µg of RNA was incubated with 200 µM dNTPs and 0.5 µg/mL Oligo (dT) 12–18 mer primers (65 °C, 5 min) followed by 1 min incubation on ice. In the second step, 2 µL 10× M-MLV reverse transcription buffer (final concentration: 1×) and M-MLV reverse transcriptase (200 units) (Sigma-Aldrich (Merck KGaA) (Darmstadt, Germany)) were added, and the volume was adjusted to 20 µL using nuclease-free water. Then, the samples were pre-warmed (RT, 1 min) and incubated at 37 °C for 50 min, and the enzymatic reaction was stopped (80 °C, 10 min). Then, 80 µL of nuclease-free water was added to achieve an estimated cDNA concentration of 10 ng/µL.

### 2.4. RT-PCR

PCR was carried out using 10 ng of cDNA with RedTaq DNA polymerase master mix (Genaxxon (Ulm, Germany) #M3029). Details regarding the primers used can be found in Section 2.6. PCR was performed using the following conditions: denaturation at 95 °C for 5 min, followed by 35 cycles of denaturation at 95 °C for 30 s, primer annealing at 60 °C for 30 s, and primer extension at 72 °C for 1 min/kb. For Pcnt, the number of cycles was increased to 39 cycles for visualization of the *Pcnt S* band. PCR products were then separated by electrophoresis using 1% agarose gel (or 2% for *Pcnt*) containing 0.2 µg/mL ethidium bromide.

### 2.5. Quantitative PCR

Quantitative PCR was conducted with a Biorad CFX Connect Real-Time PCR system (Bio-Rad Laboratories GmbH (Feldkirchen, Germany)). The reaction was carried out using 10 ng of cDNA, SYBR Green Master Mix (Life Technologies GmH (Darmstadt, Germany)), and primers at a final volume of 10 µL. The PCR was performed with three technical replicates for each sample using the following conditions: initial denaturation at 98 °C for 30 s, followed by 35 two-step cycles of denaturation at 98 °C for 15 s and primer annealing/extension, each for 30 s at 60 °C. For the analysis of real-time PCR data, the Cq values of the technical replicates were averaged (mean Cq). The mean Cq for a given target gene was normalized to the mean Cq of *GAPDH* (used as a loading control) using the following equation: Rq = 2^−Cq (Target gene)^/2^−Cq (GAPDH)^. The resulting Rq values were used to calculate and compare the relative abundance of target transcripts.

### 2.6. Primer Information

The following primers were utilized: *Mbnl1*: Fwd: 5′-GACTTGCTCACGACCAGACA-3′ and Rev: 5′-GGCTAACTGCATTTGCTGGG-3′, 234 bp; *Mbnl2*: Fwd: 5′-TGAAGCGACCTCTCGAAGC-3′ and Rev: 5′-ATCATGGGTACTGTGGGGATG-3′, 197/164 bp (two isoforms); *myogenin*: Fwd: 5′-CCAACCCAGGAGATCATTTG-3′ and Rev: 5′-CAGACAGCCCCACTTAAAA-3′, 229 bp; *AKAP6β*: Fwd: 5′-TCACCCGATCTGGGCTTTCT-3′ and Rev: 5′-GCTCCACGTGGTCGGAAATA-3′, 174 bp; nesprin-1α: Fwd: 5′-GACTGCGACT-GCGATGTCT-3′ and Rev: 5′-GGGCTTGGCCAACTCTGAC-3′, 129 bp; *Atp2a1*: Fwd: 5′-CTCATGGTCCTCAAGATCTCAC-3′ and Rev: 5′-GGGTCAGTGCCTCAGCTTTG-3′, 215/140 bp; *Ldb3*: Fwd: 5′-GGAAGATGAGGCTGATGAGTGG-3′ and Rev: 5′-TGCTGACAGTGGTAGTGCTCTTTC-3′, 574/301 bp; *Pcnt*: PcntB-Fwd 5′-CATGGC-TCTGCACAATGAAG-3′, PcntS-Fwd 5′-CAGGGCTGTTCCGTATGTTC-3′, and PcntBS-Rev 5′-GAAGTCTCCTCAGGGCATCTC-3′, 378/249 bp; *Gapdh*: Fwd: 5′-CAGAAGACTGTGGATGGCCC-3′ and Rev: 5′-AGTGTAGCCCAGGATGCCCT-3′, 279 bp. All primers were from Sigma-Aldrich (Merck KGaA) (Darmstadt, Germany).

### 2.7. Western Blotting

Protein samples were analyzed via sodium dodecyl sulfate-polyacrylamide gel electrophoresis (SDS-PAGE) followed by Western blotting and a chemiluminescence-based detection method. Cells were lysed in ice-cold RIPA buffer containing an EDTA-free protease inhibitor cocktail (cOmplete, Roche (Merck KGaA) (Darmstadt, Germany), # 11873580001). After 10 min incubation on ice, the lysates were further homogenized by passing them through a 27 G needle six times. The lysates were centrifuged at 10,000× *g* for 15 min at 4 °C, and the supernatant was carefully collected. Lysates were subjected to SDS-PAGE and transferred to a nitrocellulose membrane as previously described at 350 mA and <60 V [8]. Membranes were blocked with 5% BSA in TBS-T (1× TBS, 0.05% Tween-20) and incubated overnight at 4 °C with primary antibodies against Mbnl2 (Santa Cruz Biotechnology (Heidelberg, Germany); sc-365104, dilution 1:1000), pan-actin (Cell Signaling Technology Europe (Leiden, Netherlands), 4968S, dilution 1:1000), or FLAG (Sigma-Aldrich (Merck KGaA) (Darmstadt, Germany), F1804, dilution 1:2000). For detection, membranes were incubated with horseradish peroxidase (HRP)-conjugated secondary antibodies for 1 h at RT, followed by three 10 min washes each in TBS-T at RT and a final rinse in TBS to remove excess detergent. HRP activity was then visualized using a Chemiluminescent HRP substrate (Millipore (Merck KGaA) (Darmstadt, Germany)) according to the manufacturer’s instructions. Images were taken with an iBright 1500 (Life Technologies GmH (Darmstadt, Germany)). To ensure equal loading, the Mbnl2 signal was stripped from the membrane and subsequently re-probed with a pan-actin antibody to normalize protein expression levels. Successful stripping was confirmed by incubating the membrane with a secondary antibody alone before re-probing with the pan-actin antibody.

### 2.8. Immunofluorescence Staining and Microscopy

The primary antibodies used were the following: rabbit anti-AKAP6 (1:500, HPA048741/Sigma-Aldrich- Merck KGaA (Darmstadt, Germany)), anti-rabbit AKAP9 (1:300, HPA026109/Sigma-Aldrich Merck KGaA (Darmstadt, Germany)), mouse anti-AKAP450 (AKAP9) (1:200, 611518/BD Biosciences (Heidelberg, Germany)), rabbit anti-CDK5RAP2 (1:500, ABE236/ Merck KGaA (Darmstadt, Germany)), mouse anti-GM130 (1:500, 610823/BD Biosciences (Heidelberg, Germany)), mouse anti-PCM1 (1:250, sc-398365/Santa Cruz Biotechnology (Heidelberg, Germany)), rabbit anti-Pcnt (1:500, ab220784/Abcam (Cambridge, UK)), (all 3 min, −20 °C, methanol), rabbit anti-AKAP6 (FL100) (1:1000, gift from M.S. Kapiloff), mouse anti-myogenin (1:500, sc-12732/Santa Cruz Biotechnology (Heidelberg, Germany)), mouse anti-nesprin-1 (1:100, gift from G. Morris [16]), rabbit anti-PCM1 (1:250, sc-67204/Santa Cruz Biotechnology (Heidelberg, Germany)), rat anti-α-tubulin (1:500, ab6160/Abcam (Cambridge, UK)) (all 10 min, RT, 4% paraformaldehyde (PFA), rabbit anti-PCM1 (1:250, 10 min, RT, 4% PFA or 3 min, 20 °C, methanol, 19856-1-AP/Proteintech (Manchester, UK) ). Secondary antibodies used were: donkey anti-goat Alexa Fluor 647 (1:500, A21447), donkey anti-mouse Alexa Fluor 647 (1:500, A31571), donkey anti-mouse Alexa Fluor 594 (1:500, A21203), donkey anti-mouse Alexa Fluor 488 (1:500, A21202), donkey anti-rabbit Alexa Fluor 647 (1:500, A31573), donkey anti-rabbit Alexa Fluor 594 (1:500, A21207), donkey anti-rabbit Alexa Fluor 488 (1:500, A21206), donkey anti-rat Alexa Fluor 594 (1:500, A21209), and donkey anti-rat Alexa Fluor 488 (1:500, A21208) (all Life Technologies GmH (Darmstadt, Germany)). Cells were fixed with 4% paraformaldehyde (PFA) for 10 min at RT or ice-cold methanol for 3 min at −20 °C. PFA-fixed cells were permeabilized with 0.5% TritonX-100/PBS for 10 min at RT. Unspecific binding sites were blocked using a blocking buffer (5% BSA in 0.2% Tween-20 in PBS) for 30 min at RT. Subsequently, samples were incubated with primary antibodies diluted in blocking buffer for 90 min at RT or overnight at 4 °C in a humidified chamber. Samples were then washed three times for 10 min with washing buffer (0.1% NP40/PBS) and incubated for 1 h at RT in the dark with fluorophore-coupled secondary antibodies diluted in blocking buffer. DNA was visualized by incubating with 0.5 µg/mL DAPI solution for 5 min at RT. Millipore-grade water was used to rinse off DAPI, and samples were mounted on microscopic slides using a water-based mounting medium (Fluoromount-G Life Technologies GmH (Darmstadt, Germany)). Analysis, image acquisition, and high-resolution microscopy were performed with an LSM800 confocal laser-scanning microscope equipped with an Airyscan detector and the ZEISS Zen (Blue edition) software (Carl Zeiss AG (Oberkochen, Germany), RRID: SCR_013672) with Airyscan image processing.

### 2.9. Image Analysis

Image processing and analysis were performed using the Fiji software package (ImageJ 1.54f, NIH, US, http://fiji.sc). Raw confocal files were opened using the Bio-Formats plugin and projected into a single plane using maximum-intensity projection.

### 2.10. Detection and Counting of Nuclei

For nucleus detection, DAPI images were converted into binary images using Huang’s fuzzy thresholding method [34]. Images were then processed with the ‘watershed’ and ‘fill holes’ algorithms, followed by detecting the objects (i.e., nuclei) using the ‘Analyze Particles’ tool in the Fiji software based on empirically determined size criteria (area ≥ 70 µm^2^). The count of detected objects was treated as the number of nuclei in the respective image. That information was used to calculate, e.g., the percentage of positive nuclei. Outlines of the nuclei were then converted into a mask and overlaid onto single-channel images of the signal of interest (e.g., MTOC proteins). Positive nuclei were counted using the ‘cell counter’ plugin.

### 2.11. Bioinformatics Analysis

Bioinformatics analysis was carried out using NCBI (https://www.ncbi.nlm.nih.gov/) and the UCSC genome browser (https://genome.ucsc.edu/). The genomic sequence of *AKAP6β* was obtained from the UCSC genome browser. The alignment of sequences for different species was carried out using the Clustal O software (EMBL-EBI (European Bioinformatics Institute) (Hinxton, UK)), and, subsequently, the required sequence (zipcode sequence of ACACCC) in the 3′UTR region of the genes was analyzed for similarities and mismatches in nucleotides across species.

### 2.12. Luciferase Plasmids

A portion of the AKAP6 3′ UTR containing the zipcode sequence was amplified from the cDNA of differentiated C2C12 using the following primers: Fwd: 5′-ATTCTAGGCGATCGCTCGATAATGTAATGCCCCCAAGC-3′ and Rev: 5′-ATTTTATTGCGGCCAGCGGCAACTGTTCTCAATTACCTGAAG-3′. After amplification, the 3′ UTR fragment was cloned into the psiCHECK-2 vector (Promega GmbH (Walldorf, Germany) C8021) between the Renilla luciferase ORF (hRluc) and the synthetic poly(A) signal using NEBuilder HiFi DNA Assembly Master Mix (New England Biolabs GmbH (Frankfurt am Main, Germany), Cat# E2621L) according to the manufacturer’s instructions. The psiCHECK-2 vector was linearized via PCR using the following primers: Fwd: 5′-GCCGCTGGCCGCAATAAAATATC-3′ and Rev: 5′-ATCGAGCGATCGCCTAGAATTAC-3′.

### 2.13. Luciferase Assay

HEK293T cells (ATCC (Manassas, VA, USA)) were co-transfected with psiCHECK-2-based luciferase constructs and either Mbnl2-FLAG or an empty FLAG vector as a control using Lipofectamine LTX according to the manufacturer’s instructions. Luciferase activity was measured with the Dual-Luciferase Reporter Assay System (Promega GmbH (Walldorf, Germany) #E1910) in a 96-well plate reader luminometer (Centro XS3 LB 960, Berthold Technologies GmbH & Co. KG (Bad Wildbad, Germany), #50-6860) according to the manufacturer’s instructions. In brief, HEK293T cells were harvested 48 h after transfection in passive lysis buffer and stored at −80 °C until measurement. Activities of firefly luciferase (hluc+, internal control) and Renilla luciferase (RNA stability) were measured sequentially for each sample. The values of Renilla luciferase activity were normalized to those of firefly luciferase for each measurement.

### 2.14. Statistical Analysis

Statistical analysis and *p*-value (*p*) calculations were performed using the GraphPad Prism 5 software (GraphPad Software LLC (San Diego, CA, USA). Differences between compared groups were considered statistically significant when *p* < 0.05. Statistical significance was assessed either with one-way ANOVA followed by Bonferroni’s post hoc test to compare selected pairs of groups or Student’s *t*-test together with an F-test to assess equality of variances. In each figure legend, the utilized method is indicated as one-way ANOVA/Bonferroni or Student’s *t*-test/ F-test.

## 3. Results

### 3.1. MTOC Proteins Are Recruited in a Sequential and Gradual Manner to the Nuclear Envelope

Previously, it has been shown through gene depletion experiments that there is an inter-dependency and, thus, a hierarchy among MTOC proteins during NE-MTOC formation in skeletal muscle [6,35], as well as in cardiomyocytes and osteoclasts [8]. However, the temporal localization of MTOC proteins and the Golgi to the NE during NE-MTOC formation remains unclear. Here, the in vitro differentiation of C2C12 mouse myoblasts was used as a model system to study the timing of establishing the NE-MTOC anchor platform. This anchor platform, consisting of nesprin-1α and AKAP6, is sufficient for MTOC protein recruitment to the NE in a heterologous system [6,8] and required for NE recruitment of MTOC proteins.

Previous studies identified nesprin-1α as the anchor for AKAP6 at the NE [36]. In addition, it has been shown that AKAP6 expression at the NE is required for the perinuclear localization of MTOC proteins and the Golgi [6,8]. In order to determine whether nesprin-1α and AKAP6 appear at the same time at the NE, e.g., as a complex, or in a sequential manner, C2C12 mouse myoblasts were differentiated for two days and analyzed via co-immunofluorescence staining for nesprin-1α and AKAP6. Notably, the anti-nesprin antibody utilized here recognizes different isoforms of nesprin-1. However, it is well known that nesprin-1α is the dominant isoform in differentiated myoblasts [16]. Qualitative analysis revealed three kinds of nuclei: (i) nesprin-1α^+^/AKAP6^−^, (ii) nesprin-1α^+^/AKAP6^+^, and (iii) nesprin-1α^−^/AKAP6^−^ (*n* = 3, per *n* > 500 nuclei) (Figure 1a,b). Notably, no nucleus population that was only positive for AKAP6 was found (Figure 1b). These data suggest that both proteins are localized to the NE in a sequential manner, with nesprin-1α appearing prior to AKAP6.

Interestingly, at day 1 post-differentiation, some nesprin-1^+^/AKAP6^+^ nuclei exhibited a gradual distribution of nesprin-1 and AKAP6 across the NE, with nesprin covering a larger surface than AKAP6, indicating a gradual spreading from one of the sides (Figure 1c). Together, these data show that nesprin-1 and AKAP6 are localized at the NE in a sequential and gradual manner.

Currently, PCM1 is considered to be the first MTOC protein localized to the NE in cardiomyocytes [13]. Furthermore, AKAP6 knockdown in skeletal muscle cells has been shown to abrogate PCM1 localization at the NE [6]. To determine whether AKAP6 and PCM1 localize together to the NE or whether they are recruited in a sequential manner, C2C12 was differentiated for 1 day and co-stained for AKAP6 and PCM1. Qualitative analysis revealed that three kinds of nuclei exist: (i) AKAP6^−^/PCM1^−^, (ii) AKAP6^+^/PCM1^−^, and (iii) AKAP6^+^/PCM1^+^ (*n* = 3, per *n* > 500 nuclei) (Figure 2a,b). AKAP6^−^/PCM1^+^ nuclei were not observed. Notably, some of the AKAP6^+^/PCM1^+^ nuclei, consistently with previous reports [6], showed a partial localization of PCM1 at the NE (NE entry site), while AKAP6 uniformly covered the entire NE in these double-positive nuclei. This suggests that PCM1 translocates to the NE in a gradual fashion. Furthermore, the NE entry site of PCM1 coincides with the NE area exhibiting the most intense anti-AKAP6 staining. These data indicate that PCM1 is recruited to the NE after the NE-MTOC anchor platform is established along the entire NE.

PCM1 plays an important role in the recruitment of MTOC proteins to the centrosome [37,38,39]. Furthermore, immunoprecipitation studies have indicated a direct interaction between PCM1 and Pcnt at the centrosome, suggesting that they form a functional complex in the cells [40]. Like Pcnt, CDK5RAP2 is an essential MTOC component required for microtubule nucleation at the centrosome and the NE [41,42]. Additionally, knockdown of Pcnt affects the localization of CDK5RAP2 to the NE [35]. In the context of the NE-MTOC, it has been shown that PCM1 is required for the localization of Pcnt to the NE [35]. Consequently, it could be hypothesized that MTOC proteins such as Pcnt and CDK5RAP2 are localized to the NE in a complex with PCM1. In order to validate this hypothesis, C2C12 cells were differentiated for 1 day and co-stained for PCM1 and Pcnt or CDK5RAP2 (Figure 3). The analysis of PCM1^+^/Pcnt^+^ and PCM1^+^/CDK5RAP2^+^ nuclei revealed that the NE distribution patterns of PCM1 and Pcnt or CDK5RAP2 were significantly different. While the staining for PCM1 and Pcnt or CDK5RAP2 overlapped, showing the highest signal intensity at the nuclear region proximal to the centrosome (PCM1^+^ cloud, yellow asterisk), PCM1 occupied a larger domain compared to Pcnt (*n* = 3, per *n* > 500 nuclei) and CDK5RAP2 (*n* = 3, per *n* > 300 nuclei) (Figure 3a–d). In contrast, the analysis of Pcnt^+^/CDK5RAP2^+^ nuclei revealed that Pcnt and CDK5RAP2 occupied the same domain in all analyzed nuclei (Figure 3e). These data suggest that MTOC proteins are recruited from a centrosomal localization and that Pcnt and CDK5RAP2 might be recruited together. However, MTOC proteins do not appear to be recruited to the NE in a complex with PCM1.

### 3.2. AKAP9 Localizes with the Golgi to the NE After the Recruitment of MTOC Proteins

AKAP9 is known to localize to the centrosome, NE, and Golgi and is critical for microtubule nucleation. In addition, it is required to maintain the Golgi’s integrity [43,44]. Furthermore, it has been shown in cardiomyocytes that AKAP9 binds to the NE-MTOC via interaction with AKAP6 at the NE [8], while its NE localization is independent of PCM1 or Pcnt [35]. To understand the recruitment of AKAP9 to the NE, C2C12 cells were stained 1 day post-differentiation for AKAP9 together with PCM1, Pcnt, or CDK5RAP2. Unlike PCM1, Pcnt, or CDK5RAP2, which uniformly covered the NE, AKAP9 was restricted to specific NE regions (Figure 4a), suggesting that its recruitment to the NE occurs after Pcnt and CDK5RAP2 in an independent manner. Interestingly, after 3 days of differentiation, there remained a significant number of PCM1^+^ nuclei where AKAP9 was either not detected or localized in a gradient around the nuclei (Figure 4b,c). In Figure 4b, we illustrate different patterns of AKAP9 localization at the NE with numbered examples: (1) PCM1^+^ nuclei without detectable AKAP9, (2) PCM1^+^ nuclei with AKAP9 localized in a gradient at one side of the nucleus, (3) PCM1^+^ nuclei where AKAP9 covered a larger portion of the NE but was still unevenly distributed, and (4) PCM1^+^ nuclei with complete coverage of AKAP9 across the entire NE. These examples highlight the diverse distribution patterns of AKAP9 observed at different stages of its recruitment to the NE, illustrating the sequential (PCM1 and then AKAP9) and gradual (AKAP9) recruitment of MTOC proteins.

To assess the localization pattern of the Golgi during NE-MTOC formation, cells were stained for the Golgi protein GM130 in combination with PCM1, Pcnt, or CDK5RAP2 (Appendix A). These data indicated that the Golgi and AKAP9 are recruited to the NE in a similar fashion. Analysis of co-staining confirmed the assumption that AKAP9 and GM130 co-localize at the NE (Figure 4d), indicating that AKAP9 is recruited with the Golgi to the NE.

### 3.3. Mbnl2 Is Required for NE-MTOC Formation in Skeletal Muscle Cells

In recent years, several components of NE-MTOCs in striated muscle cells have been identified. In addition, recently, the interdependencies of these components have been elucidated in cardiomyocytes [8] and skeletal muscle cells [6,35]. Yet, the factors that control the switch from a centrosomal to a non-centrosomal MTOC remain elusive. Here, the role of Mbnl as a regulator of alternative splicing in muscle cells was investigated, as it has been shown that different isoforms of some MTOC proteins, such as Pcnt and ninein, are utilized in centrosomal and non-centrosomal MTOCs [3]. In order to characterize the skeletal muscle differentiation system utilized here in regard to Mbnl expression, mRNA expression of *Mbnl1* and *Mbnl2* in C2C12 cells was analyzed via RT-PCR before induction and at days 1, 2, 3, and 5 post-induction of differentiation. The Ct-value for *Mbnl1* was in the range of 25 to 26 at day 0, whereas it was from 28 to 29 for *Mnbl2*. This suggests that Mbnl1 has markedly higher expression in proliferating C2C12 myoblasts than Mbnl2. To validate the induction of differentiation, myogenin expression was monitored. As shown in Appendix A, the levels of *myogenin* increased constantly from day 1 to day 5. The analysis of *Mbnl1* expression revealed that *Mbnl1* mRNA expression was significantly upregulated in 5-day-differentiated C2C12 cells compared to proliferating C2C12 myoblasts (day 0) (Appendix A). In contrast, *Mbnl2* mRNA expression levels were relatively constant, showing no significant difference on day 5 compared to day 0 but a small, significant transient downregulation on day 2 (Appendix A). These data suggest that Mbnl1 might be more relevant than Mbnl2 for NE-MTOC formation in C2C12 cells.

To determine whether Mbnl1 and/or Mbnl2 are required for NE-MTOC formation, siRNA-mediated knockdown of Mbnl1 and/or Mbnl2 (henceforth termed as siMbnl1, siMbnl2, and siMbnl1/2) was established in C2C12 cells, and non-targeting siRNA (siControl) was used as a control. Proliferating C2C12 cells were transfected with siRNAs, induced to differentiate 24 h later, and analyzed 48 h post-induction of differentiation (equaling 72 h post-transfection). Analysis of *Mbnl1* and *Mbnl2* mRNA expression upon siMbnl1 or siMbnl2 transfection resulted in a downregulation of ~87% for *Mbnl1* and ~88% for *Mbnl2* (Appendix A). Notably, siMbnl1-mediated downregulation of *Mbnl1* was accompanied by an approximately fourfold increase in *Mbnl2* mRNA levels (Appendix A). In contrast, siMbnl2 transfection had no effect on *Mbnl1* mRNA levels. Co-transfection with siMbnl1 and siMbnl2 resulted in a downregulation of ~89% for *Mbnl1* and ~67% for *Mbnl2* mRNA, respectively. Western blot analysis for Mbnl2 levels validated the qPCR data (Appendix A). WB for Mbnl1 could not be performed due to the lack of a specific anti-Mbnl1 antibody. Pan-actin was used as a loading control for Western blotting (Appendix A). To test whether the achieved knockdown of Mbnl1 impairs alternative splicing, two exons whose inclusion or exclusion is known to be controlled by Mbnl1 were analyzed. Mbnl1 promotes exon 11 skipping of *Ldb3* and exon 22 inclusion of *Atp2a1* [45]. In siControl-treated C2C12 cells, roughly equal levels of both alternatively spliced isoforms (included and skipped) for *Ldb3* and *Atp2a1* were detected (Appendix A). In contrast, in siMbnl1-treated cells, as well as siMbnl1-/siMbnl2-treated cells, the *Ldb3* isoform including exon 11, as well as the *Atp2a1* isoform excluding exon 22, was upregulated (Appendix A). This type of analysis was not performed for siMbnl2-treated cells, as no *Mbnl2* target gene is known to be expressed in skeletal muscle cells. These data indicate that the established knockdown strategy for Mbnl1 and Mbnl2 is effective and allows the determination of whether Mbnl1 and/or Mbnl2 play a role in NE-MTOC formation.

To determine whether Mbnl1 and/or Mbnl2 knockdown affects NE-MTOC formation in C2C12 cells, PCM1 expression at the NE was analyzed via immunofluorescence staining. In addition, the nuclear expression of myogenin, which is known to control NE-MTOC in myoblasts and to induce NE-MTOC formation in fibroblasts when ectopically expressed [6], was monitored to assess whether Mbnl1 and/or Mbnl2 knockdown affects early skeletal muscle differentiation. The analysis of anti-PCM1/anti-myogenin co-staining in 2-day-differentiated C2C12 cells revealed three kinds of nuclei: (i) myogenin^−^/PCM1^−^, (ii) myogenin^+^/PCM1^−^, and (iii) myogenin^+^/PCM1^+^ (Figure 5a,b). No nuclei positive for PCM1 alone were observed. Quantitative analysis revealed that in siControl-treated cell cultures, as well as siMbnl1- and/or siMbnl2-treated cell cultures, ~30% of the nuclei were myogenin^+^ (Figure 5c). This suggests that the knockdown of Mbnl1 and/or Mbnl2 does not affect early muscle differentiation. In contrast, quantitative analysis of myogenin^+^/PCM1^+^ nuclei showed a significant difference between the different treatment groups (Figure 5b). In siControl- and siMbnl1-treated cell populations, ~40% of the myogenin^+^ nuclei were PCM1^+^. In contrast, in siMbnl2-treated cell populations, the number of myogenin^+^/PCM1^+^ nuclei was significantly reduced to ~15%. Similarly, double knockdown of Mbnl1 and Mbnl2 resulted in a reduction to ~23% myogenin^+^/PCM1^+^ nuclei. Taken together, these data suggest that not Mbnl1 but Mbnl2 is required for the efficient establishment of NE-MTOCs in skeletal muscle cells.

### 3.4. Mbnl2 Regulates the Expression Levels of AKAP6β

The inefficient recruitment of PCM1 to the NE upon Mbnl2 knockdown might result from effects on AKAP6, the adaptor for MTOC proteins, or nesprin-1α, the nuclear anchor of AKAP6. To test this, we first analyzed nesprin-1. Immunostaining analysis revealed that the number of nesprin-1α^+^ nuclei in control and Mbnl2-depleted C2C12 cells did not differ significantly (Figure 6a,b). In addition, qPCR analysis demonstrated that *nesprin-1α* mRNA levels were not significantly different between siControl- and siMbnl2-treated cells two days post-induction of differentiation (Figure 6c). In contrast, the number of AKAP6+ nuclei in differentiated myogenin^+^ cells was reduced from ~66% in siControl-treated cells to ~36% upon Mbnl2 depletion (Figure 6d,e). Furthermore, mRNA levels of *AKAP6β*, the predominant isoform in striated muscle cells [15], were reduced by ~50% in siMbnl2-treated cells compared to siControl-treated cells (Figure 6f). Importantly, *myogenin* mRNA levels showed no significant difference, confirming comparable differentiation efficiency in the different treatment groups (Figure 6g). These data suggest that Mbnl2 knockdown reduces the expression of AKAP6, impairing NE-MTOC formation in differentiated C2C12 cells.

Mbnl proteins are RNA-binding proteins and are known to control pre-mRNA splicing, as well as RNA stability and RNA transport [25,26,46]. The observation that Mbnl2 depletion results in reduced *AKAP6β* mRNA levels and that nesprin-1α and AKAP6β arise from alternative internal promoters and not from alternative splicing suggests that Mbnl2 regulates the *AKAP6β* mRNA stability. As it has been previously shown that Mbnl2 binds to a so-called zipcode sequence, ACACCC, in the 3′UTR of other mRNAs, such as *β-actin* and *integrin-α3* pre-mRNAs [46], we investigated whether this zipcode sequence exists in the 3′UTR of *AKAP6β* pre-mRNA and whether the 3′UTR around the zipcode sequence ACACCC is conserved. For this purpose, the 3′UTR of mouse (12871 nucleotides, ENSMUST00000095737.4), human (12911 nucleotides, ENST00000280979.9), and rat (6263 nucleotides, NM_022618.1) *AKAP6* was investigated. This analysis showed that the 3′UTR of mouse, human, and rat *AKAP6* contains two (starting position: 1538 and 5428), one (starting position: 3804), and one (starting position: 1616) zipcode sequences, respectively (Appendix A).

The Clustal O alignment revealed the highest conservation around the mouse zipcode sequence at position 1538 (Appendix A). While mouse and rat sequences were identical in regard to 59 nucleotides (mouse starting position: 1527), only two nucleotides in the zipcode sequence were different in the human sequence (ACACCC vs. ACATTC). Finally, the second mouse zipcode sequence at position 5428 was also partially conserved over a stretch of 40 nucleotides (Appendix A). However, both the human and rat sequences differed in a single position in the zipcode sequence and six surrounding positions. When the sequences related to the two extended mouse zipcode sequences were compared, nucleotides at nine positions outside the zipcode sequence were conserved (Appendix A). Based on the high conservation of the mouse zipcode sequence at position 1538 with a potential ACATTC zipcode sequence in humans, the existence of this sequence in the different 3′UTRs was assessed. This analysis showed that the 3′UTR of mouse, human, and rat *AKAP6* contains eight, five, and two zipcode sequences, respectively (Appendix A). This analysis revealed another small conserved 25-nucleotide-long sequence in the 3′UTR of *AKAP6* (mouse starting position: 5679), which differs only in the human sequence by one nucleotide (insertion of a nucleotide in front of the potential zipcode sequence ACATTC) (Appendix A). For all other zipcode sequence positions in the 3′UTRs, no obvious conservation between the mouse, human, and rat was observed. Taken together, these data indicate that while some zipcode sequences in the *AKAP6β* 3′UTR show partial conservation, the overall evidence does not support the hypothesis that Mbnl2 regulates *AKAP6β* mRNA stability via direct interaction with its 3′UTR.

To experimentally assess whether Mbnl2 directly regulates the stability of *AKAP6β* mRNA via its zipcode, the *AKAP6β* 3′UTR sequence, including its zipcode elements, was inserted into a reporter construct downstream of the Renilla luciferase (hRluc) ORF (Appendix A). Co-expression of Mbnl2-FLAG with this construct in HEK293T cells did not lead to significant changes in relative luciferase activity when compared to the empty FLAG control (Appendix A). These results indicate that there is no substantial stabilization of the *AKAP6β* 3′UTR in the presence of Mbnl2-FLAG. Consistently with bioinformatic predictions, these findings suggest that Mbnl2 does not regulate *AKAP6β* mRNA stability through direct interaction with its 3′UTR.

### 3.5. Mbnl2 Depletion Affects Pcnt Isoform Expression

In cardiomyocytes, expression of the Pcnt isoform Pcnt S is associated with NE-MTOC formation [13,47]. Pcnt B is the predominant isoform at the centrosome in proliferating cells; the shorter Pcnt S isoform is expressed predominantly at the NE in postnatal cardiomyocytes. The mechanism controlling the expression of Pcnt B and Pcnt S is unknown. In order to assess whether Mbnl proteins affect Pcnt isoform expression, RT-PCR analyses with primers specific for *Pcnt B* and *S* were carried out post-Mbnl-depletion in C2C12 cells. The percentage of *Pcnt S* with respect to total *Pcnt* in siMbnl1-, siMbnl2-, and siMbnl1/2-treated cells was determined and normalized to the percentage of *Pcnt S* in siControl-treated cells. These data showed that Mbnl2 depletion resulted in a significant reduction in the percentage of *Pcnt S*, while Mbnl1 depletion had no significant effect (Figure 7).

Depletion of Mbnl1 and Mbnl2 together showed a clear reduction by 54%, which was, however, not significant (Figure 7). This might be due to a less efficient Mbnl2 knockdown in siMbnl1/2-treated cells, as siMbnl1 treatment results in Mbnl2 upregulation (Appendix A). Note that the total amount of *Pcnt* was not significantly affected by Mbnl2 knockdown (101 ± 12%). These data indicate that Mbnl2 plays a role in regulating *Pcnt* isoform expression, particularly in maintaining the relative abundance of the PcntS isoform.

## 4. Discussion

We conclude based on co-immunofluorescence staining assays that nesprin-1α is expressed prior to AKAP6 NE localization during C2C12 myoblast differentiation. This is in agreement with previous studies where the overexpression of a truncation mutant of nesprin-1α that cannot be targeted to the NE affects the perinuclear localization of AKAP6 [36] or the release of AKAP6 from the NE in SYNE1^−/−^ human myoblasts [6]. Interestingly, although AKAP6 never shows a partial NE localization, it often appears to form a gradient around the NE, with a higher signal intensity proximal to the centrosome. This AKAP6 gradient coincides with the gradual recruitment of MTOC proteins from a centrosomal localization, as concluded from the existence of nuclei covered only partially with several different MTOC proteins at the NE facing the centrosome.

We further conclude based on the comparison of the coverage of the NE by the different MTOC proteins that PCM1 is the first protein recruited to the NE in a gradual manner with an NE entry point at a position facing the centrosome. Notably, PCM1 appears to be an integral part of the NE-MTOC, forming its inner layer together with AKAP6, while it is a major part of the centrosomal satellites and is only transiently associated with the centrosomal MTOC [48]. Subsequently, Pcnt and CDK5RAP2 are recruited to the NE, also in a gradual fashion. Notably, PCM1 knockdown abrogates the NE localization of Pcnt, and Pcnt knockdown abolishes CDK5RAP2 NE localization [35,49]. As PCM1 always occupies a larger NE domain than Pcnt and CDK5RAP2, it appears that Pcnt and CDK5RAP2 are not transported to the NE in a complex with PCM1. Whether Pcnt and CDK5RAP2 can directly interact with each other and form a complex prior to NE localization remains unknown. Similarly, it remains to be investigated whether (a) MTOC proteins are translocated from the centrosome to the NE, (b) centriolar satellites are rerouted to transport MTOC proteins to the NE instead of the centrosome, or (c) a totally different transport mechanism exists.

After PCM1, Pcnt, and CDK5RAP2 are detected at the NE, AKAP9 is localized to the NE. In contrast to the aforementioned MTOC proteins, AKAP9 NE localization appears to be unaffected by PCM1 or Pcnt knockdown [35]. However, the number of AKAP9-positive nuclei in skeletal muscle cells is reduced when AKAP6 is depleted [6]. It has been further shown that NE localization of Pcnt and AKAP9 is dependent on the direct interaction of their PACT domain with the SR1 domain of AKAP6 [8]. The data presented here show that AKAP9 exhibits a patchy distribution in skeletal muscle cells and co-localizes with the Golgi, as previously observed in cardiomyocytes [8]. This is congruent with earlier reports that AKAP9 associates not only with the centrosome but also with the Golgi [50]. Localization of AKAP9 to the Golgi depends on the cis-Golgi protein GM130, and microtubule nucleation from the Golgi is largely dependent on AKAP9 [8,51]. Thus, AKAP9 tethers the Golgi by binding GM130 to the NE via binding of its PACT domain to AKAP6 [8].

Taken together, our data suggest that the anchor platform consisting of nesprin-1α and AKAP6 is established before the recruitment of MTOC proteins. AKAP6 is followed by PCM1. Subsequently, Pcnt and CDK5RAP2 recruitment occurs, and, finally, AKAP9 and Golgi recruitment takes place (Figure 8).

While several components of NE-MTOCs and their interdependencies in striated muscle cells have been identified, it remains elusive what factors/mechanisms control the switch from a centrosomal to a non-centrosomal MTOC in mammals. Here, we show that Mbnl2 is required for proper NE-MTOC formation. While C2C12 differentiation was associated with an increased *Mbnl1* expression, *Mbnl2* expression changed only slightly, indicating a major role for Mbnl1 in NE-MTOC formation. However, siRNA-mediated depletion experiments showed that Mbnl2 is required for proper NE-MTOC formation, without affecting skeletal muscle differentiation based on myogenin expression. Interestingly, siRNA-mediated depletion of Mbnl1 resulted in an approximately fourfold increase in *Mbnl2* levels, suggesting that Mbnl2 might partially rescue the loss of Mbnl1.

Mbnl proteins are known to affect the expression of their targets by either modulating RNA stability and/or splicing. Here, we show that Mbnl2 affects NE-MTOC formation through both mechanisms. First, our data show that Mbnl2 depletion affects NE-MTOC formation by reducing *AKAP6* mRNA levels. Whether this is due to the known function of Mbnl2 acting as an mRNA stabilizing factor by directly binding to the 3′UTR of target pre-mRNAs is questionable. Adereth et al. (2005) identified a so-called zipcode sequence, ACACCC, in the 3′UTR region of *integrin α3* [46], which is required for the transport of integrin α3 from the cell body to adhesion plaques in lung carcinoma cell lines. This sequence was initially described to be required for the transport of mRNAs encoding members of the actin family [52]. Mbnl2 protein was shown to directly interact with this sequence via RNA immunoprecipitation assays [46]. Removal of the zipcode sequence from the 3′UTR region of *integrin α3* abolished the interaction with Mbnl2 and its transport.

Our bioinformatics analysis identified a conserved zipcode sequence (ACACCC) in the 3′UTR of mouse *AKAP6β*, with conservation in rats and only partial conservation in humans (ACATTC). While the surrounding RNA sequence is highly conserved across species, our results provide limited support for a direct interaction between Mbnl2 and the *AKAP6β* 3′UTR. To further investigate the role of this region in *AKAP6β* mRNA stability, we performed luciferase reporter assays. These experiments revealed no significant changes in luciferase activity between cells co-expressing Mbnl2-FLAG and the control, suggesting that Mbnl2 does not substantially stabilize the *AKAP6β* 3′UTR. Further experiments are needed to clarify whether Mbnl2 binds to this conserved region and how this interaction, if present, affects *AKAP6β* mRNA stability and localization. Second, our data show that Mbnl2 controls the expression levels of Pcnt isoforms. However, it remains unclear how Mbnl2 contributes to altered Pcnt S levels. Notably, Mbnl2 levels remained relatively constant during differentiation, and, thus, its activity does not appear to be controlled on an expression level. Future experiments are needed to determine whether Mbnl2 activity depends on the regulation of co-factors or whether Mbnl2 can directly bind to the *Pcnt S* mRNA stabilizing it.

## 5. Conclusions

The results shown in this work demonstrate that the NE-MTOC is formed by the recruitment of its components in a sequential and gradual manner to the NE and that Mbnl2 is required for proper NE-MTOC formation. Taken together, our results shed light on how mammals post-transcriptionally control the switch from a centrosomal MTOC to an NE-MTOC and identify Mbnl2 as a novel modulator of ncMTOCs in skeletal muscle cells.

## Figures and Tables

**Figure 1 cells-14-00237-f001:**
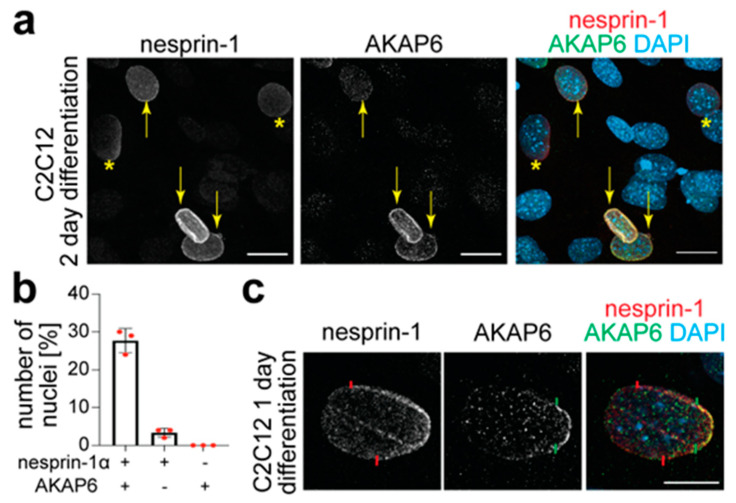
Nesprin-1α is observed at the NE before AKAP6. (**a**) Immunofluorescence analysis of C2C12 differentiated for 2 days showing co-staining for nesprin-1 (red) and AKAP6 (green). Note that nesprin-1^+^/AKAP6^+^ nuclei are indicated by yellow arrows and nesprin-1^+^/AKAP6^−^ nuclei are indicated by yellow asterisks. Scale bar: 20 µm. (**b**) Quantitative analysis of (**a**). *n* = 3 per *n* > 500 nuclei. Data: mean ± SD. (**c**) Immunofluorescence analysis of C2C12 differentiated for 1 day showing a gradient in the signal intensity of nesprin-1 and AKAP6. The edges of the surface covered by each protein are indicated with a colored line: red: nesprin-1; green: AKAP6. Scale bar: 10 µm.

**Figure 2 cells-14-00237-f002:**
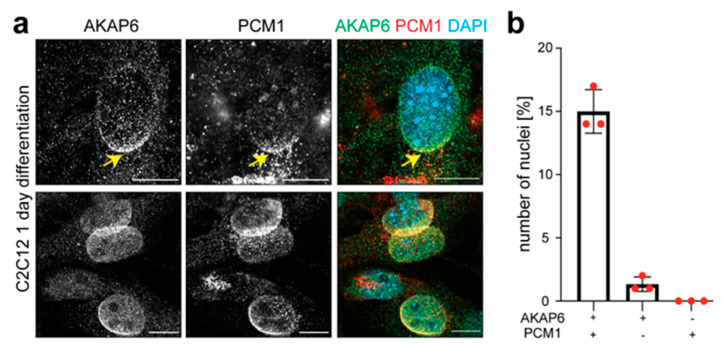
PCM1 is recruited to the NE after the anchor platform is established. (**a**) Immunofluorescence analysis of C2C12 differentiated for 1 day showing two examples of co-staining for AKAP6 (green) and PCM1 (red). The yellow arrow represents the higher signal intensity of AKAP6 at the NE where PCM1 is recruited. Scale bar: 10 µm. (**b**) Quantitative analysis of (**a**). *n* = 3 per *n* > 500 nuclei. Data: mean ± SD.

**Figure 3 cells-14-00237-f003:**
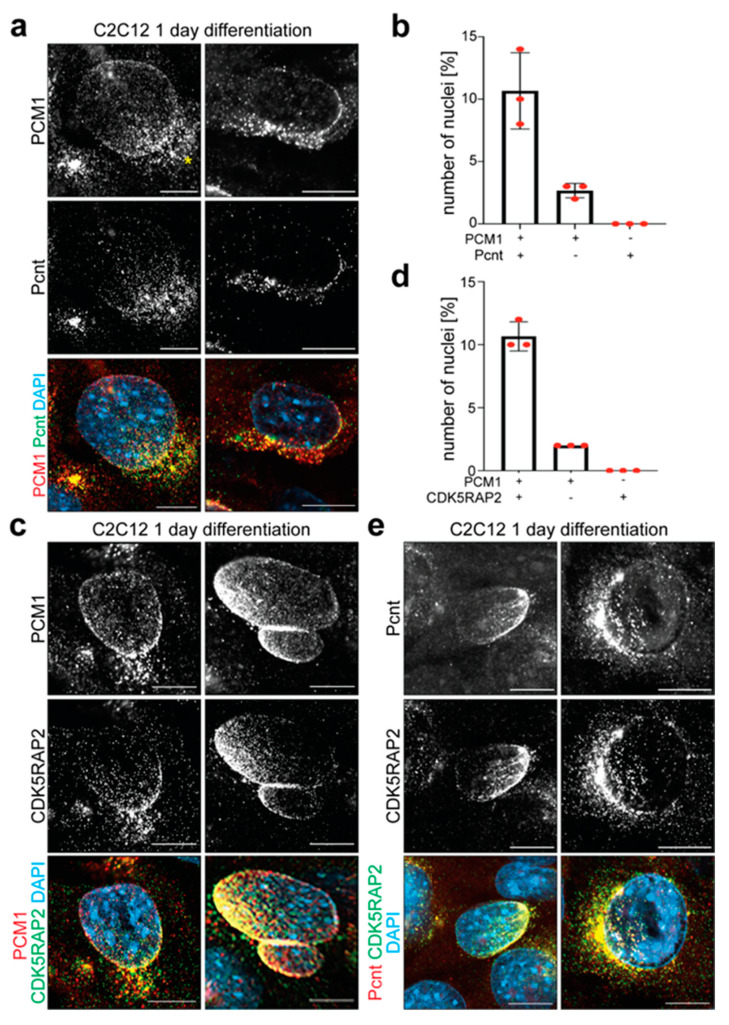
MTOC proteins do not appear to be localized to the NE in a complex with PCM1. (**a**,**c**) Immunofluorescence analysis of C2C12 differentiated for 1 day showing two examples of co-staining of PCM1 (red) with Pcnt (green) (**a**) or CDKRAP2 (green) (**c**). Yellow asterisk: nuclear region proximal to the centrosome. (**b**,**d**) Quantitative analysis of (**a**,**c**). *n* = 3 per *n* > 500 (**b**) or >300 (**d**) nuclei. Data: mean ± SD. (**e**) Immunofluorescence analysis of C2C12 differentiated for 1 day showing two examples of co-staining of Pcnt (red) and CDK5RAP2 (green). Scale bars: 10 µm.

**Figure 4 cells-14-00237-f004:**
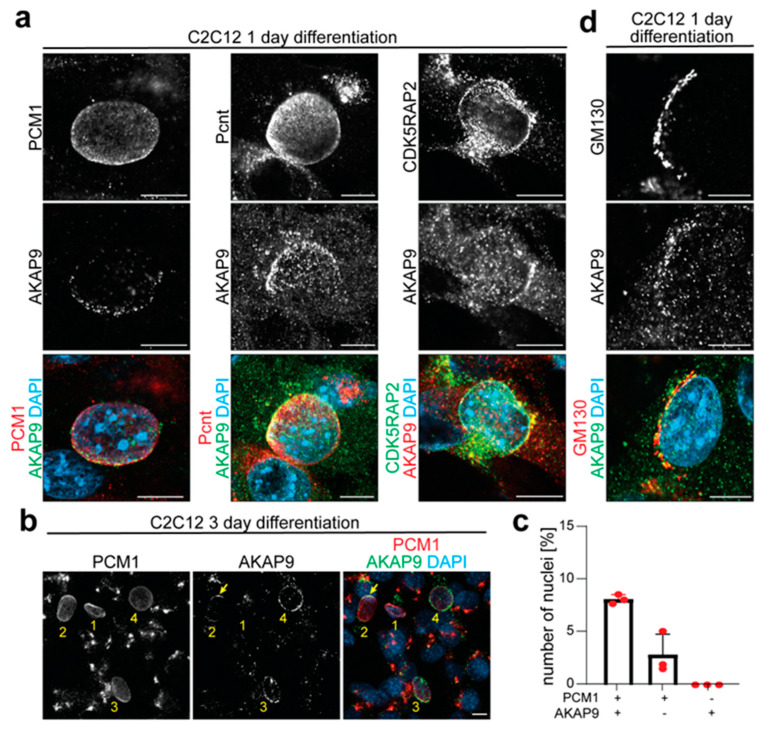
AKAP9 and GM130 co-localize at the NE. (**a**) Immunofluorescence analysis of C2C12 differentiated for 1 day showing co-staining of AKAP9 with the MTOC proteins PCM1 (red), Pcnt (red), or CDK5RAP2 (green). (**b**) Immunofluorescence analysis of C2C12 differentiated for 3 days illustrating different examples of the gradient localization of AKAP9 (green) at the NE of PCM1^+^ nuclei. Categories include the following: (1) PCM1^+^ nuclei without AKAP9, (2) localization at one side of the nucleus (arrow), (3) greater coverage of the NE than in (3), and (4) full coverage. (**c**) Quantitative analysis of (**b**), where PCM1^+^/AKAP9^+^ represents categories 2–4. *n* = 3 per *n* > 200 nuclei. (**d**) Immunofluorescence analysis of C2C12 differentiated for 1 day showing co-staining of AKAP9 (green) with the Golgi marker GM130 (red). Scale bar: 10 µm.

**Figure 5 cells-14-00237-f005:**
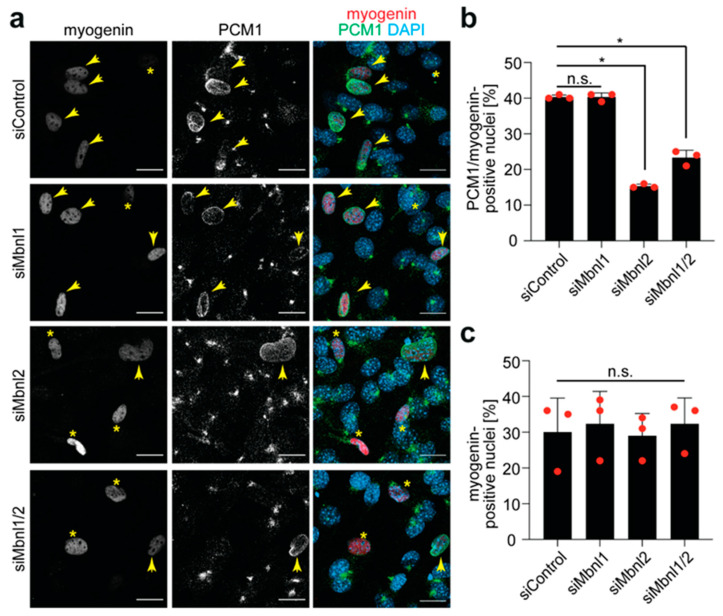
PCM1 is mislocalized in siMbnl2- and siMbnl1/2-treated C2C12 cells. (**a**) Immunofluorescence analysis of siControl- and siMbnl-treated C2C12 cells showing co-staining for myogenin (red) and PCM1 (green). Yellow arrows: myogenin^+^/PCM1^+^ nuclei. Yellow asterisks: myogenin^+^/PCM1^−^ nuclei. Scale bar: 20 µm. (**b**,**c**) Quantitative analysis of the number of PCM1^+^ nuclei in myogenin^+^ cells and total number of myogenin^+^ cells (%). Data are shown as individual biological replicates with the mean ± SD. *n* = 3, n.s.: not significant (*p* > 0.05), *: *p* < 0.05. (**b**,**c**): One-way ANOVA/Bonferroni.

**Figure 6 cells-14-00237-f006:**
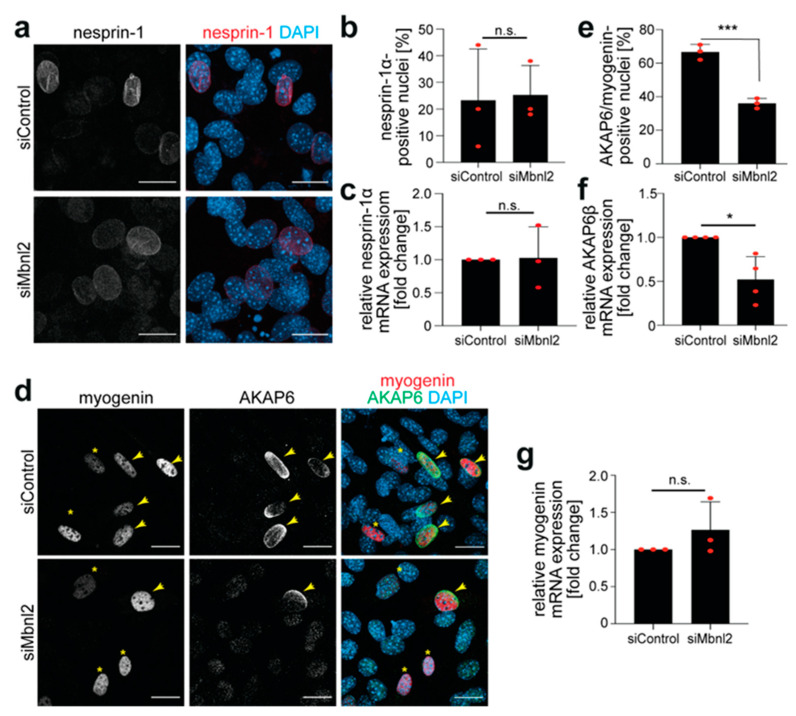
Mbnl2 regulates the expression levels of AKAP6β but not those of nesprin-1α. (**a**) Immunofluorescence analysis of siControl- and siMbnl2-treated C2C12 cells showing nesprin-1 staining (red). Scale bar = 20 µm. (**b**) Quantitative analysis of the number of nesprin-1α+ nuclei in siControl- and siMbnl2-treated C2C12 cells. (**c**) qPCR analysis of the level of *nesprin-1α* mRNA expression in siControl- and siMbnl2-treated C2C12 cells. (**d**) Immunofluorescence analysis of siControl- and siMbnl2-treated C2C12 showing co-staining of myogenin (red) and AKAP6 (green). Yellow asterisks: myogenin^+^/AKAP6^−^ nuclei. Scale bar = 20 µm. (**e**) Quantification of AKAP6/myogenin-positive nuclei in control and treated cells. (**f**,**g**) qPCR analysis of *AKAP6β* and *myogenin* mRNA levels in siControl- and siMbnl2-treated C2C12 cells ((**f**): *n* = 4). Data are shown as individual biological replicates with the mean ± SD. *n* = 3 if not stated otherwise. n.s.: not significant (*p* > 0.05), *: *p* < 0.05, ***: *p* < 0.001. (**b**,**c**,**e**–**g**): Student’s *t*-test/F-test.

**Figure 7 cells-14-00237-f007:**
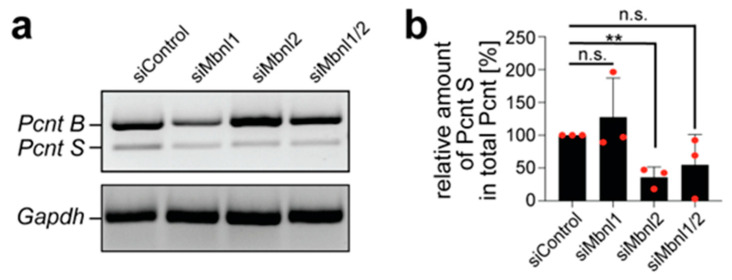
Mbnl2 depletion affects the level of *Pcnt S* in differentiated C2C12 cells. (**a**) Agarose gel showing the RNA levels of *Pcnt B* and *Pcnt S* 72 h post-siRNA-transfection as indicated. *Gapdh* is used as a loading control. (**b**) Quantification of the relative amount of *Pcnt S* present in the total *Pcnt* [%], normalized to siControl. Data are shown as individual biological replicates with the mean ± SD. *n* = 3. n.s.: not significant (*p* > 0.05), **: *p* < 0.01. (**b**): One-way ANOVA/Bonferroni.

**Figure 8 cells-14-00237-f008:**
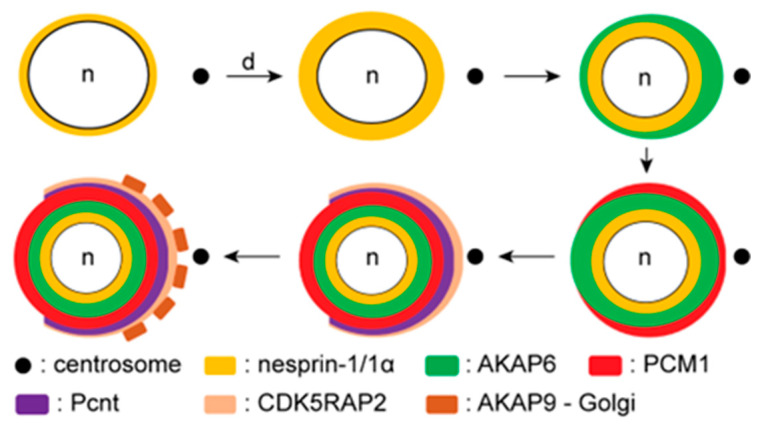
Schematic model of the spatiotemporal NE-MTOC component recruitment during skeletal muscle differentiation. In proliferating myoblasts, the outer nuclear membrane of the nucleus (n) expresses the giant isoform of nesprin-1. As differentiation (d) begins, nesprin-1α is specifically upregulated in skeletal muscle, and its expression appears before AKAP6’s localization to the NE. AKAP6 shows a gradient with a higher signal intensity proximal to the centrosome. Once the NE-MTOC anchor platform is established, PCM1 is the first MTOC protein to be recruited to the NE, followed by Pcnt and CDK5RAP2. Pcnt and CDK5RAP2 cover the same nuclear domain, in contrast to PCM1, which has a higher nuclear coverage area. Finally, AKAP9 localizes to the NE via the Golgi, whereby both of the components show a patchy distribution around the NE during the early stages of differentiation.

## Data Availability

The data are all included in the article and Appendix A. No omics data were generated.

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
