# Peer review of "NE-MTOC Formation in Skeletal Muscle Is Mbnl2-Dependent and Occurs in a Sequential and Gradual Manner"

_cells, 2025, doi:10.3390/cells14040237_

Round 1
Reviewer 1 Report
Comments and Suggestions for Authors
Comment: In this work the authors investigated the study of NE-MTOC formation by the in vitro differentiation of C2C12 mouse myoblasts as a model system. They found based on longitudinal co-immunofluorescence staining analyses that MTOC proteins are recruited in a sequential and gradual manner to the NE. AKAP9 localizes with the Golgi to the NE after the recruitment of MTOC proteins. Moreover, siRNA-mediated depletion experiments revealed that Mbnl2 is required for proper NE-MTOC formation by regulating the expression levels of AKAP6β. Also, they showed, Mbnl2 depletion affects Pcnt isoform expression. Based on the results a model was illuminated how mammals post-transcriptionally control the switch from a centrosomal MTOC to an NE-MTOC and identify Mbnl2 as a novel modulator of ncMTOCs in skeletal muscle cells.
Major comment:
In general, the claims made here are well-supported by the data. However, clear explanation of the sequential and gradual appearance of proteins to NE formation is somewhat confusing. Although this is addressed in the in vitro immunofluorescence assays, I am still left with some questions over the extent of this phenomenon. Can the authors provide some experimental evidence to support the sequential appearance of the proteins by clearly mentioning the definition of sequential and gradual manner. Also, how the proteins are sequentially appeared depending on the time sequence of cell differentiation should be clearly presented such as in Fig. 1, Fig. 4 to clarify the point.
Minor comments:
In Fig. 1, 2, 3, 4, the letter (d) mentioned is missing in the figures which create large confusions.
What is the final concentration of siRNA knockdown in case of siMbnl1/2?
In description of Fig 4, 4a (line 364), 4b (line 370) were mentioned but missing in the figure.
In Fig. 5, unify the letters (A, B, C to a, b, c).
Author Response
We thank the reviewers for their in-depth review providing a number of constructive comments to improve our manuscript. In the revised manuscript we have addressed all the raised points by changing the text and/or providing new data. Please note, changes based on reviewer comments are highlighted in yellow, changes requested by the editor are highlighted in green.
Reviewer 1
Comment: In this work the authors investigated the study of NE-MTOC formation by the in vitro differentiation of C2C12 mouse myoblasts as a model system. They found based on longitudinal co-immunofluorescence staining analyses that MTOC proteins are recruited in a sequential and gradual manner to the NE. AKAP9 localizes with the Golgi to the NE after the recruitment of MTOC proteins. Moreover, siRNA-mediated depletion experiments revealed that Mbnl2 is required for proper NE-MTOC formation by regulating the expression levels of AKAP6β. Also, they showed, Mbnl2 depletion affects Pcnt isoform expression. Based on the results a model was illuminated how mammals post-transcriptionally control the switch from a centrosomal MTOC to an NE-MTOC and identify Mbnl2 as a novel modulator of ncMTOCs in skeletal muscle cells.
Major comment:
In general, the claims made here are well-supported by the data. However, clear explanation of the sequential and gradual appearance of proteins to NE formation is somewhat confusing. Although this is addressed in the in vitro immunofluorescence assays, I am still left with some questions over the extent of this phenomenon. Can the authors provide some experimental evidence to support the sequential appearance of the proteins by clearly mentioning the definition of sequential and gradual manner. Also, how the proteins are sequentially appeared depending on the time sequence of cell differentiation should be clearly presented such as in Fig. 1, Fig. 4 to clarify the point.
We thank the reviewer for pointing out that we have not clearly defined the difference between sequential and gradual appearance of the MTOC proteins. To clarify this point, we have made the following modifications to the manuscript:
- Introduction, page 3, line 113-117: “In this study, we used in vitro differentiation of C2C12 mouse myoblasts to study NE-MTOC formation and showed that MTOC proteins are recruited to the NE in a sequential and gradual manner, meaning the appearance of two proteins one after the other at the NE or the appearance of a protein at the NE that gradually, over time covers the NE, respectively.”.
- Results, page 7, line 307-315: “Notably, no nuclei population only positive for AKAP6 was found (Figure 1b). These data suggest that both proteins are localized to the NE in a sequential manner, with nesprin-1α appearing prior to AKAP6. Interestingly, at day 1 post-differentiation, some nesprin-1+/AKAP6+ nuclei exhibited a gradual distribution of nesprin-1 and AKAP6 across the NE, with nesprin covering a larger surface than AKAP6, indicating a gradual spreading from one of the sides (Figure 1c). Together, these data show that nesprin-1 and AKAP6 are localized at the NE in a sequential and gradual manner.”
- Results, page 7, lines 321-323: “Immunofluorescence analysis of C2C12 differentiated for 1 day showing a gradient in the signal intensity of nesprin-1 and AKAP6. The edges of the surface covered by each protein is indicated with a colored line: red: nesprin-1; green: AKAP6.”
- Results, page 8 to 9, lines 376-385. “Interestingly, after 3 days of differentiation, there remained a significant number of PCM1+ nuclei where AKAP9 was either not detected or localized in a gradient around the nuclei (Figure 4b,c). In Figure 4b, we illustrate different patterns of AKAP9 localization at the NE with numbered examples: (1) PCM1+ nuclei without detectable AKAP9, (2) PCM1+ nuclei with AKAP9 localized in a gradient at one side of the nucleus, (3) PCM1+ nuclei where AKAP9 covered a larger portion of the NE but was still unevenly distributed, and (4) PCM1+ nuclei with complete coverage of AKAP9 across the entire NE. These examples highlight the diverse distribution patterns of AKAP9 observed at different stages of its recruitment to the NE illustrating the sequential (PCM1 and then AKAP9) and gradual (AKAP9) recruitment of MTOC proteins.”
- Figure 4, page 10, line 402-406: We have included new panels (b and c): “(b) Immunofluorescence analysis of C2C12 differentiated for 3 days illustrating different examples of gradient localization of AKAP9 at the NE of PCM1+ Categories include: (1) PCM1+ nuclei without AKAP9, (2) localization at one side of the nuclei (arrow), (3) greater coverage of the NE than in (3), and (4) full coverage. (c) Quantitative analysis of (b), where PCM1+/AKAP9+ represents categories 2-4. n = 3, per n > 200 nuclei.”
Minor comments:
In Fig. 1, 2, 3, 4, the letter (d) mentioned is missing in the figures which create large confusions.
Thank you for pointing this out. Actually, the “(d)” is the abbreviation for “day” in the figure referring to “1 d differentiation”. As this appears to cause confusion, we have omitted this abbreviation and spelled “d” in the figures out and removed the abbreviation from the figure legends.
What is the final concentration of siRNA knockdown in case of siMbnl1/2?
The final concentration of siMbnl1/2 is 40 nM (20 nM for siMbnl1 and 20 nM for siMbnl2). This information has been added to the Material and Methods section on page 3, lines 128-129.
In description of Fig 4, 4a (line 364), 4b (line 370) were mentioned but missing in the figure.
Thank you for spotting this. By rearranging Figure 4 for optimal representation during manuscript preparation, we forgot to include “a” and “b”. Figure 4 has been corrected accordingly, and two more panels have been added to this figure.
In Fig. 5, unify the letters (A, B, C to a, b, c).
Thank you for this comment. The figure has been corrected accordingly.
Reviewer 2 Report
Comments and Suggestions for Authors
In this study, the authors aimed to determine whether NE-MTOC formation modulates the function of differentiated cells. To achieve this, they investigated how NE-MTOC components are assembled and whether post-transcriptional mechanisms regulate NE-MTOC formation. Using in vitro differentiation of C2C12 mouse myoblasts combined with co-immunofluorescence staining, the authors demonstrated that MTOC proteins (PCM-1, Pcnt, and CDK5RAP2) are recruited to the NE in a sequential and gradual manner. AKAP9 was found to localize with the Golgi to the NE after the recruitment of MTOC proteins. Furthermore, siRNA experiments revealed that Mbnl2, a highly enriched muscle-specific splicing factor, regulates NE-MTOC formation. This regulation likely occurs through Mbnl2's role in modulating AKAP6b expression and Pcnt splicing, suggesting a post-transcriptional mechanism that controls the transition from a centrosomal MTOC to an NE-MTOC. These findings identify Mbnl2 as a novel modulator of ncMTOCs in skeletal muscle cells.
Although the data presented is generally robust, the study is mainly descriptive. Several important questions and points of clarification must be addressed prior to publication, as outlined below:
-Given that nesprins undergo extensive alternative splicing, further investigation is required to determine if Mbnl2 regulates the expression of both nesprin-1 and AKAP6. Additionally, it is worth exploring whether Mbnl1 knockdown via siRNA affects Pcnt levels, considering the potential compensatory role of Mbnl2 during Mbnl1 depletion and its relatively stable levels during differentiation.
-Conduct RNA-binding assays to elucidate the mechanisms by which Mbnl2 regulates AKAP6b expression.
-Determine if nesprin-1 knockdown impacts the localization of Pcnt and CDK5RAP2, as the authors suggest that these proteins are transported to the NE in a PCM-1-independent manner.
-Most data were collected at days 1 and 2 (primarily myoblast stage). Additional experiments on day 3, when myotubes begin to form, would provide greater insight into NE-MTOC dynamics during differentiation.
-Figure 1: Include nesprin-1 staining in Fig. 1c and perform Western blotting for both nesprin-1 and AKAP6 from days 1 to 3.
-Page 10, Line 427: Clarify whether the reference is to fibroblasts or myoblasts.
-Figure 6: Perform WB for nesprin-1, AKAP6, and myogenin. For Fig. 6a, include myogenin staining, and for Fig. 6b, present analysis of nesprin-1a/myogenin-positive nuclei as done for Fig. 6e.
-Figure 7: Provide Western blot data for Fig. 7.
-Page 15, Line 613: Add a comma between “...its transport” and “Our bioinformatics...”.
-Supplementary Figure S2: Perform WB for figures 2a and 2b (Mbnl1 WB is optional if antibodies are unavailable). Provide statistical analysis for panels S2e and S2f.
-Use a consistent black background with white bands for all PCR images to distinguish them from WB images.
Author Response
We thank the reviewers for their in-depth review providing a number of constructive comments to improve our manuscript. In the revised manuscript we have addressed all the raised points by changing the text and/or providing new data. Please note, changes based on reviewer comments are highlighted in yellow, changes requested by the editor are highlighted in green.
Reviewer 2
In this study, the authors aimed to determine whether NE-MTOC formation modulates the function of differentiated cells. To achieve this, they investigated how NE-MTOC components are assembled and whether post-transcriptional mechanisms regulate NE-MTOC formation. Using in vitro differentiation of C2C12 mouse myoblasts combined with co-immunofluorescence staining, the authors demonstrated that MTOC proteins (PCM-1, Pcnt, and CDK5RAP2) are recruited to the NE in a sequential and gradual manner. AKAP9 was found to localize with the Golgi to the NE after the recruitment of MTOC proteins. Furthermore, siRNA experiments revealed that Mbnl2, a highly enriched muscle-specific splicing factor, regulates NE-MTOC formation. This regulation likely occurs through Mbnl2's role in modulating AKAP6b expression and Pcnt splicing, suggesting a post-transcriptional mechanism that controls the transition from a centrosomal MTOC to an NE-MTOC. These findings identify Mbnl2 as a novel modulator of ncMTOCs in skeletal muscle cells.
Although the data presented is generally robust, the study is mainly descriptive. Several important questions and points of clarification must be addressed prior to publication, as outlined below:
-Given that nesprins undergo extensive alternative splicing, further investigation is required to determine if Mbnl2 regulates the expression of both nesprin-1 and AKAP6.
Please note:
- Nesprin-1α and AKAP6β arise from alternative internal promoters and not from classical alternative splicing. This information has been included on page 12, lines 498-499.
- As shown in Figure 6c, Mbnl2 knockdown has no effect on nesprin-1α expression.
- However, the expression of AKAP6β was significantly affected (Figure 6f). To determine whether Mbnl2 regulates AKAP6β mRNA stability, we have performed a luciferase reporter assay. For this purpose, the AKAP6β 3’UTR, including its zipcode elements, was cloned into a reporter construct downstream of the Renilla luciferase ORF (hRluc) and co-transfected with Mbnl2-FLAG in HEK293T. Co-expression of Mbnl2-FLAG with the AKAP6β 3’UTR fused to the 3’ of hRluc construct in HEK293T cells did not lead to significant changes in relative luciferase activity when compared to the empty FLAG control. These results indicate that there is no substantial stabilization of the AKAP6β 3’UTR in the presence of Mbnl2-FLAG and confirms the bioinformatic analysis indicating that Mbnl2 does not regulate AKAP6β mRNA stability via direct interaction with its 3’UTR. The Luciferase data have been added as follows:
- New Figure: Figure S5
- Results, page 14, line 543-551: “To experimentally assess whether Mbnl2 directly regulates the stability of AKAP6β mRNA via its zipcode, the AKAP6β 3’UTR sequence, including its zipcode elements, was inserted into a reporter construct downstream of the Renilla luciferase (hRluc) ORF (Figure S5a). Co-expression of Mbnl2-FLAG with this construct in HEK293T cells did not lead to significant changes in relative luciferase activity when compared to the empty FLAG control (Figure S5b). These results indicate that there is no substantial stabilization of the AKAP6β 3’UTR in the presence of Mbnl2-FLAG. Consistent with bioinformatic predictions, these findings suggest that Mbnl2 does not regulate AKAP6β mRNA stability through direct interaction with its 3’UTR.”
- Methods, page 6, lines 257-279.
- Discussion, page 16, lines 658-661: “To further investigate the role of this region in AKAP6β mRNA stability, we performed luciferase reporter assays. These experiments revealed no significant changes in luciferase activity between cells co-expressing Mbnl2-FLAG and the control, suggesting that Mbnl2 does not substantially stabilize the AKAP6β 3’UTR.”
Further mechanistical studies are beyond the scope of this manucript.
- Additionally, it is worth exploring whether Mbnl1 knockdown via siRNA affects Pcnt levels, considering the potential compensatory role of Mbnl2 during Mbnl1 depletion and its relatively stable levels during differentiation.
The effect of Mbnl1 and/or Mbnl2 knockdown on Pcnt B and PcntS mRNA expression has been shown in Figure 7 and described on page 14, lines 557-562.
-Conduct RNA-binding assays to elucidate the mechanisms by which Mbnl2 regulates AKAP6b expression.
See response to the first comment.
-Determine if nesprin-1 knockdown impacts the localization of Pcnt and CDK5RAP2, as the authors suggest that these proteins are transported to the NE in a PCM-1-independent manner.
We thank the reviewer for pointing this out. Actually, we had initially stated that “Pcnt and CDK5RAP2 might be transported to the NE in a PCM1-independent manner” and then realized that this phrase might be misleading and thus corrected the manuscript. Yet, we only had changed the text in the Results (page 8, line 363-366: “These data suggest that MTOC proteins are recruited from a centrosomal localization and that Pcnt and CDK5RAP2 might be recruited together. However, MTOC proteins appear not to be recruited to the NE in a complex with PCM1.”) and missed to correct the statement in the Discussion (original manuscript lines 555-557: “As PCM1 always occupies a larger NE domain than Pcnt and CDK5RAP2, it appears that Pcnt and CDK5RAP2 are transported to the NE in a PCM1-independent manner.”)
Note, we and others have previously demonstrated that the recruitment of Pcnt to the NE is PCM1-dependent (Becker et al., Elife 2021; Gimpel et al. Curr Biol 2017). Here, we provide evidence that Pcnt and CDK5RAP2 are not transported as a complex with PCM1 to the NE but in a sequential manner, meaning first PCM1 and then Pcnt and CDK5RAP2. In addition, it is established that nesprin-1 depletion abrogates the localization of AKAP6, MTOC proteins (PCM1, Pcnt, AKAP9), and GM130 to the NE, as nesprin-1 serves as the anchor for the MTOC at the NE (Vergarajauregui et al., Elife 2020; Becker et al., Elife 2021; Gimpel et al., Curr Biol 2017; Espigat-Georger et al., J Cell Sci 2016).
To address this point, we have now changed the Discussion, which reads now (page 15, lines 594-596): “As PCM1 always occupies a larger NE domain than Pcnt and CDK5RAP2, it appears that Pcnt and CDK5RAP2 are not transported to the NE in a complex with PCM1”.
-Most data were collected at days 1 and 2 (primarily myoblast stage). Additional experiments on day 3, when myotubes begin to form, would provide greater insight into NE-MTOC dynamics during differentiation.
We regretfully disagree with this statement. It is well established that the NE-MTOC is established in myoblast before myotube formation (Fant et al., Plos One 2009; Srsen et al., BMC Cell Biol 2009). Thus, in our opinion the analysis at day 3 won’t provide any additional information on NE-MTOC dynamics during differentiation.
-Figure 1: Include nesprin-1 staining in Fig. 1c and perform Western blotting for both nesprin-1 and AKAP6 from days 1 to 3.
We have now included nesprin-1 staining in Figure 1c. To better visualize the sequential and gradual recruitment of nesprin-1 and AKAP6 to the NE, we have provided data at day 1 of differentiation (Figure 1c). The Results section has been changed as follows (page 7, lines 311-315) “Interestingly, at day 1 post-differentiation, some nesprin-1+/AKAP6+ nuclei exhibited a gradual distribution of nesprin-1 and AKAP6 across the NE, with nesprin covering a larger surface than AKAP6, indicating a gradual spreading from one of the sides (Figure 1c). Together, these data show that nesprin-1 and AKAP6 are localized at the NE in a sequential and gradual manner.”. In addition, the figure legend of Figure 1 has been changed to “Immunofluorescence analysis of C2C12 differentiated for 1 day showing a gradient in the signal intensity of nesprin-1 and AKAP6. The edges of the surface covered by each protein is indicated with a colored line: red: nesprin-1; green: AKAP6.”
Regarding western blotting analyses, we would like to emphasize that global expression levels of nesprin-1 and AKAP6 are not informative for studying NE-MTOC dynamics during differentiation. Western blotting does not provide information about expression levels in individual cells or the specific localization of nesprin-1α within the nuclear envelope. Therefore, immunofluorescence-based analyses were performed to address these aspects. Furthermore, the global protein expression levels of nesprin-1α during C2C12 differentiation have already been described (Gimpel et al. Curr Biol 2017). Analysis of AKAP6 presents additional technical challenges. Even in the adult heart, which exhibits the highest AKAP6 expression, its detection by standard western blotting is very challenging. AKAP6 is a large protein (220 kDa) and requires immunoprecipitation for detection. Since the number of differentiated myoblasts is relatively low, the levels of AKAP6 in these cells are too low for reliable western blot detection, making this approach unfeasible.
-Page 10, Line 427: Clarify whether the reference is to fibroblasts or myoblasts.
This refers to Becker et al. (Elife 2021). Here we have shown that depletion of myogenin abrogates NE-MTOC formation in myoblasts, but induces NE-MTOC when ectopically expressed in fibroblasts. The text has been modified accordingly (Results, page 11, lines 456-457).
-Figure 6: Perform WB for nesprin-1, AKAP6, and myogenin. For Fig. 6a, include myogenin staining, and for Fig. 6b, present analysis of nesprin-1a/myogenin-positive nuclei as done for Fig. 6e.
Regarding western blotting analyses see our response above. Regarding myogenin staining and analysis for Figure 6, we unfortunately cannot perform the suggested experiments due to antibody incompatibility, since both antibodies (nesprin-1a and myogenin) are raised in mouse.
-Figure 7: Provide Western blot data for Fig. 7.
Unfortunately, there is no specific antibodies available that are suitable for western blotting analysis of Pcnt. We have attempted this several times in the recent years and exchanged our experience with collaborators. However, the available antibodies generate a large number of non-specific bands, making reliable analysis impossible.
-Page 15, Line 613: Add a comma between “...its transport” and “Our bioinformatics...”.
The manuscript has been changed by adding a full stop, “.”, after transport (Discussion, page 16, line 640).
-Supplementary Figure S2: Perform WB for figures 2a and 2b (Mbnl1 WB is optional if antibodies are unavailable). Provide statistical analysis for panels S2e and S2f.
In our opinion, western blot analysis is not particularly insightful for this study, as it does not provide information on protein expression levels in individual cells or the specific localization of the investigated protein within the cell. Similarly, mRNA expression data, while valuable, does not address these aspects, which is why we have included these data into the supplementary information. Instead, we have chosen to perform immunofluorescence-based analyses, which allow us to directly assess protein localization and expression in individual cells. Furthermore, myogenin and nesprin-1α have already been extensively studied during C2C12 differentiation. As mentioned above, analysis of AKAP6β protein expression is technically challenging and not feasible. Therefore, we have focused on providing protein expression data for Mbnl2 in this study. Lastly, it is important to note that no specific anti-Mbnl1 antibodies are currently available for reliable detection.
As the expression levels of Mbnl1 and Mbnl2 are both important, considering compensation, and no anti-Mbnl1 antibody is available, we have quantified the effect of siRNA-mediated knockdown on Mbnl1 and Mbnl2 expression by RT-PCR (Figure S2 c,d). These data clearly prove that the chosen siRNAs efficiently decrease Mbnl1/2 expression. The western blot analysis of Mbnl2 expression represents qualitative data to confirm that Mbnl2 knockdown correlates to strongly reduced Mbnl2 protein expression.
-Use a consistent black background with white bands for all PCR images to distinguish them from WB images.
We have evaluated the appearance and interpretability of the PCR images with black background and white bands. In our opinion, the interpretation of the results is clearer for the reader in the current format. In addition, there are now only two western blots presented in the revised manuscript. To avoid any confusion, we have now clearly identified and stated that these two images (Figure S2, e and f) represent western blots (in the figures as well as the figure legends) to clearly differentiate them from the PCR images.
Round 2
Reviewer 1 Report
Comments and Suggestions for Authors
I suggest accepting the revised manuscript in the current form.
Author Response
We thank the reviewer for the positive response.
Reviewer 2 Report
Comments and Suggestions for Authors
The authors answered most of the questions and the manuscript has been improved.
For the Fig. 6b, it was suggested to include myogenin staining and present the graph as nesprin-1a/myogenin-positive nuclei as done for Fig. 6e. Authors mentioned they cannot perform the suggested experiments due to antibody incompatibility (both antibodies, nesprin-1a and myogenin, are raised in mouse). There are good antibodies against myogenin are available via Thermo Fisher (eg. rabbit polyclonal antibody PA5-116750, mouse monoclonal MA5-11486 ,14-5643-82), which would further support their conclusion.
Author Response
Comments 1 "For the Fig. 6b, it was suggested to include myogenin staining and present the graph as nesprin-1a/myogenin-positive nuclei as done for Fig. 6e. Authors mentioned they cannot perform the suggested experiments due to antibody incompatibility (both antibodies, nesprin-1a and myogenin, are raised in mouse). There are good antibodies against myogenin are available via Thermo Fisher (eg. rabbit polyclonal antibody PA5-116750, mouse monoclonal MA5-11486 ,14-5643-82), which would further support their conclusion."
Response 1: We thank the reviewer for the positive response. We appreciate the reviewer’s suggestion to include myogenin staining and present the graph as nesprin-1α/myogenin-positive nuclei, as done for Figure 6e for AKAP6. While we initially mentioned antibody incompatibility as a limitation (we meant antibodies available in our lab), we thank the reviewer for the recommendation of alternative antibodies for myogenin, such as the rabbit polyclonal antibody (PA5-116750).
While the suggested experiment appears feasible, it will be time consuming to repeat all experiments and establishing the staining procedures. In our opinion, these experiments will not provide additional information for the following reasons:
- Figure 6b shows that Mbnl2 depletion does not significantly alter the number of nesprin-1α-positive nuclei.
- Figure 6c shows that nesprin-1α mRNA levels remain consistent regardless of treatment conditions.
- Figure 5c shows that Mbnl2 depletion does not significantly affect the number of myogenin-positive cells.
- Figure 6g shows that myogenin mRNA levels were not significantly affected by Mbnl2 depletion.
Taken together, we believe the existing data already provide sufficient support for our conclusions and thus decided not to perform these additional experiments; also to fulfill the 10 day revision limit.
Round 3
Reviewer 2 Report
Comments and Suggestions for Authors
In the future, it would be beneficial to obtain the antibody and conduct the experiments before manuscript submission, as it is readily available.